# Advances in Diagnostic Imaging for Cardiac Sarcoidosis

**DOI:** 10.3390/jcm10245808

**Published:** 2021-12-11

**Authors:** Osamu Manabe, Noriko Oyama-Manabe, Tadao Aikawa, Satonori Tsuneta, Nagara Tamaki

**Affiliations:** 1Department of Radiology, Jichi Medical University Saitama Medical Center, Saitama 330-8503, Japan; omanabe@jichi.ac.jp (O.M.); aikawatadao@gmail.com (T.A.); 2Department of Diagnostic and Interventional Radiology, Hokkaido University Hospital, Sapporo 060-8648, Japan; subsky.s1t2.7he9ia4n.uguisu@gmail.com; 3Department of Radiology, Kyoto Prefectural University of Medicine, Kyoto 602-8566, Japan; natamaki@koto.kpu-m.ac.jp

**Keywords:** cardiac sarcoidosis, ^18^F-fluorodeoxyglucose, cardiac magnetic resonance (CMR), positron emission tomography (PET)

## Abstract

Sarcoidosis is a systemic granulomatous disease of unknown etiology, and its clinical presentation depends on the affected organ. Cardiac sarcoidosis (CS) is one of the leading causes of death among patients with sarcoidosis. The clinical manifestations of CS are heterogeneous, and range from asymptomatic to life-threatening arrhythmias and progressive heart failure due to the extent and location of granulomatous inflammation in the myocardium. Advances in imaging techniques have played a pivotal role in the evaluation of CS because histological diagnoses obtained by myocardial biopsy tend to have lower sensitivity. The diagnosis of CS is challenging, and several approaches, notably those using positron emission tomography and cardiac magnetic resonance imaging (MRI), have been reported. Delayed-enhanced computed tomography (CT) may also be used for diagnosing CS in patients with MRI-incompatible devices and allows acceptable evaluation of myocardial hyperenhancement in such patients. This article reviews the advances in imaging techniques for the evaluation of CS.

## 1. Introduction

Sarcoidosis is an inflammatory granulomatous disease of unknown etiology that can affect any organ, including the lungs, lymphatic system, skin, eyes, and heart [1]. Cardiac sarcoidosis (CS) is clinically recognized in approximately 5% of patients with systemic sarcoidosis, whereas CS has been recorded in at least 25% of autopsy or imaging studies [2,3]. The incidence and prevalence of CS vary substantially over different geographical regions and ethnic groups. An autopsy series of patients with systemic sarcoidosis showed that approximately 27% from the United States and 80% from Japan were revealed to have CS [4]. CS is a potentially life-threatening condition that can lead to conduction disturbances, such as atrial and ventricular arrhythmias and heart failure. Therefore, screening for CS, including medical interviews, physical examinations, and electrocardiography (ECG), is recommended for patients newly diagnosed with sarcoidosis [1]. 

The presentation of CS varies from asymptomatic to chest pain, dyspnea, syncope, palpitations, and sudden cardiac death. Endomyocardial biopsies are highly specific for the diagnosis of CS, but their sensitivity is not very high; therefore, findings on cardiac magnetic resonance imaging (CMR) or positron emission tomography (PET) with ^18^F-fluorodeoxyglucose (FDG) are used for clinical diagnosis [4]. Recent studies have demonstrated that advanced cardiac imaging with CMR or PET can be useful not only for diagnosis but also for monitoring and predicting adverse cardiovascular events. In addition to visual evaluations, various quantitative evaluations have been proposed, and new imaging and analysis methods are also attracting attention. This article reviews the advances in imaging techniques for the evaluation of CS.

## 2. Guidelines for the Diagnosis of CS

Several guidelines and diagnostic criteria have been suggested for the diagnosis of CS. One of the initial guidelines was proposed by the Japanese Ministry of Health and Welfare (JMHW) in 1993 and revised in 2006, which was subsequently modified in 2015 by the Japanese Society of Sarcoidosis and Other Granulomatous Disorders (JSSOG) (Table 1) [5,6,7]. The Japanese Circulation Society (JCS) expanded the JMHW/JSSOG 2015 guidelines, including a pathway for the clinical diagnosis of isolated CS in 2016 [8]. The Heart Rhythm Society (HRS) published an expert consensus statement in 2014 (Table 2) [9]. The diagnostic pathways for CS are primarily divided into histological and clinical branches. Histopathological diagnosis is required to reveal noncaseating granulomas from endomyocardial biopsy (EMB) or surgical resection of the heart. However, the role of biopsy is limited due to its low sensitivity rate of approximately 20% to 30% [10], which may be attributed to the patchy distribution. Clinical diagnosis requires concordance among electrocardiography, echocardiography, and imaging findings, including late gadolinium enhancement (LGE) on CMR and FDG-PET (Figure 1). The advantages and disadvantages of FDG-PET and MRI are summarized in Table 3. The diagnostic criteria for CS have been updated according to the technological development and cumulating of the information about CS.

## 3. FDG-PET

FDG-PET is now routinely used to assess the infiltration of sarcoidosis in the myocardium. FDG is a glucose analog, and its uptake is associated with the expression of glucose transporters (GLUTs). Increased uptake of FDG in inflammatory diseases, including sarcoidosis, can be attributed to the activation of inflammatory cells, such as macrophages, lymphocytes, and granulocytes, with increased levels of GLUT.

The pitfall of using FDG-PET for evaluation of CS is that living myocardial cells also use glucose as their energy source. Therefore, physiological myocardial FDG uptake often makes it difficult to detect active CS lesions. Patients with suspected CS require adequate preparation, such as prolonged fasting, a low-carbohydrate diet, and/or a high-fat, high-protein diet to suppress the false-positive association with physiological myocardial FDG uptake (Figure 2) (Table 4) [13,14,15]. Sample foods for dietary preparation prior to FDG-PET scan are well summarized in the Japanese Society of Nuclear Cardiology recommendation [13]. Multivariate analysis of patients without cardiac disease, including the factors such as body mass index, fasting blood sugar, gender, age, and long fasting, demonstrated that only long fasting was independently associated with the suppression of physiological FDG uptake [16]. Measurement of serum free fatty acid levels at the injection of FDG can be used to determine whether the uptake was physiological [17]. Typical patterns of FDG accumulation in CS are focal and focal-on-diffuse in the myocardium, which often corresponds to a decrease in myocardial perfusion [13]. FDG-PET with an adequate preparation protocol is ideal for detecting active lesions and assessing the response to anti-inflammatory therapy in patients with CS [18]. The inter-observer interpretation of cardiac FDG uptake patterns was improved by proper dietary preparation [19]. A meta-analysis of data collected from 891 patients showed that PET or PET/CT had a pooled sensitivity of 84% and a pooled specificity of 83% for diagnosing CS [20]. In particular, the specificity of FDG in the detection of CS varies and is relatively low compared with its sensitivity. Possible explanations for this include nonspecific physiological uptake, uptake of other myocardial diseases such as ischemic disease and hypertrophic cardiomyopathy, and early-stage sarcoid lesions in patients who do not meet the diagnostic criteria for CS. Cardiac PET enhances the prognostic assessment of patients with suspected CS. Blankstein et al. showed that patients with abnormal FDG uptake and focal perfusion defects had a higher risk of death or ventricular tachycardia in their retrospective study of 118 patients with known or suspected CS [21]. 

Focal FDG uptake in the right ventricle (RV) is more reliable for the diagnosis of CS because physiological FDG uptake in the RV is less frequently observed than in the LV [22]. FDG uptake by the RV is associated with positive endomyocardial biopsy findings in patients with CS [23]. Assessment of RV FDG uptake provides useful information for diagnosing CS and predicting cardiac events. CS with FDG uptake in the RV is associated with subsequent death or ventricular tachycardia (VT) [21].

### 3.1. Quantitative Analysis of FDG-PET Findings

PET enables quantitative estimation of disease activity in vivo. Semi-quantitative methods, SUV measurement, and especially SUVmax evaluations, are most frequently used to assess the degree of FDG uptake. SUV is the ratio of the tissue radioactivity concentration divided by the injected dose and body weight. When fasting for over 18 h followed a low-carbohydrate diet, quantification of myocardial SUVmax provided higher diagnostic accuracy for diagnosing CS in comparison with visual analysis [24]. The pre-treatment FDG uptake index, defined as the product of the SUVmax and the number of LV segments with abnormal uptake, was a predictor of the response to immunosuppressive therapy in patients with CS [25].

SUVmax is a simple measurement, and the SUVmax value reflects only the value of a single voxel that is hampered by physical, biological, and technical processes and does not reflect the metabolism of the entire target lesion. Cardiac metabolic volume (CMV) and cardiac metabolic activity (CMA) have emerged as volume-based parameters, which are mainly useful for assessing active CS lesions [17,26]. CMV is a measurement of the volume with a high metabolism in the myocardium within a given boundary determined by a threshold, such as the liver uptake and the blood pool SUV (Figure 3). On the other hand, CMA is defined as the product of the mean SUV and the CMV. Osborne et al. reported that a reduction in SUVmax and CMV after immunotherapy was associated with improvement in the left ventricular ejection fraction (LVEF) [27]. Ahmadian et al. reported that CMA was greater in patients with lower LVEF and that an elevated CMA preceded adverse clinical events in their study of CS [26].

An alternative method to assess CS that employs quantitative evaluation of the heterogeneity of myocardial FDG uptake has been proposed by several groups [28,29,30]. Quantitative measures of the coefficient of variation (COV) of myocardial FDG uptake provide an incremental diagnostic and prognostic advantage in CS patients. Tahara et al. reported that the standard deviation and COV were significantly higher in patients with CS than in control participants, sarcoidosis patients without cardiac lesions, and dilated cardiomyopathy patients [28]. Recently, the textural features of FDG-PET images of CS have been studied for their diagnostic and prognostic value [31,32]. The quantitative assessment of FDG uptake heterogeneity may be reflected in the inhomogeneous formation of noncaseating granulomas in the myocardium of patients with CS. Novel semiconductor PET/CT with high temporal and spatial resolution enables a more detailed distribution evaluation of CS [33,34].

### 3.2. The Other Tracers

Although FDG-PET provides high diagnostic value for the detection of active CS, physiological accumulation may occasionally make it difficult to determine whether the finding is pathological or not. Several other PET tracers without physiological myocardial uptake have been used to evaluate CS. Somatostatin receptors (SSTRs) are known to be present in inflammatory cells, including granuloma macrophages, epithelioid cells, and giant cells [35]. Therefore, ^68^Ga-DOTA-Tyr-octreotide (DOTATOC), which is a PET tracer that targets SSTRs, is used to assess sarcoidosis [36,37] (Table 5). ^18^F-fluoromisonidazole (FMISO) is a PET tracer that accumulates in hypoxic tissues. Increased FMISO uptake has been reported in patients with suspected CS [38]. A small prospective study demonstrated that FMISO-PET has the potential to detect hypoxic lesions of CS without physiological myocardial uptake [39]. 3′-Deoxy-3-[^18^F]-fluorothymidine (FLT) is another PET tracer used to evaluate cellular proliferative activity. Although FLT uptake is significantly lower than that of FDG, FLT PET/CT can detect cardiac and extracardiac sarcoidosis [40]. Martineau et al. examined the relationship between FLT, FDG, and perfusion PET imaging in patients with CS [41]. They hypothesized that FLT might identify areas likely to develop myocardial scarring due to the strong correlation between FLT uptake and perfusion defects. 

## 4. CMR

### 4.1. Late Gadolinium Enhancement

Gadolinium-based contrast agents are distributed within the extracellular space of the myocardium, and their distribution is influenced by different regional conditions such as edema, scar tissue, fibrosis, and necrosis. Evaluation of LGE on CMR is a basic technique used for tissue characterization, particularly fibrosis and scar formation in the myocardium, and is an essential tool for assessing CS (Figure 4). Although the presence of LGE is a non-specific finding, the appearance of mid-wall and/or epicardial LGE sparing the subendocardium in the basal heart, particularly the septum and lateral wall, may increase the specificity for the diagnosis of CS [42,43,44,45]. A meta-analysis of data collected from 649 participants with histologically proven extra-cardiac sarcoidosis and/or with a suspicion of CS showed that CMR had a pooled sensitivity of 93% and a pooled specificity of 85% for diagnosing CS [46]. The findings showing contiguous extension to the right ventricle was also a typical pattern for CS [47]. The presence of LGE on CMR is known to be an independent predictor of all-cause mortality and adverse cardiovascular events in sarcoidosis patients [48,49]. Conversely, the prognosis is favorable for patients without abnormal LGE on CMR. A quantitative method to estimate the LGE percentage of LV mass has been proposed, and a large extent of LGE correlates with a high incidence of adverse outcomes in patients with CS [50]. According to a retrospective cohort study by Kazmirczak et al., patients with LGE extent > 5.7% had high annualized event rates [51].

Nevertheless, the presence of LGE cannot distinguish between active inflammation and chronic fibrotic changes. In this regard, a high signal intensity on T2-weighted images reflects active inflammation due to the indication of edema [52]. Tonegawa-Kuji et al. analyzed T2-weighted short-tau-inversion-recovery black-blood imaging data by measuring the myocardium-to-spleen ratio and concluded that semi-quantitative analysis was useful for the detection of active CS [53]. Currently, the development of novel techniques for CMR imaging, such as T1 and T2 mapping, has the potential to allow assessment of CS. The ECV fraction was calculated using pre- and post-enhanced T1 mapping. T1 and T2 native values are directly estimated by mapping sequences that allow reliable tissue characterization of the myocardium. Previous reports have shown that T1 and T2 values are longer in patients with systemic sarcoidosis than in healthy individuals [54,55]. T1 and T2 mapping may also be useful for recognizing cardiac involvement in patients with systemic sarcoidosis. T1 and T2 values can also be used for monitoring disease activity because these values are reduced by anti-inflammatory treatment [54].

### 4.2. Integrated PET/MRI

The hybrid PET/MRI system is a novel instrument that allows simultaneous data acquisition by combining functional, structural, and molecular imaging in a single scan [56]. Several studies have evaluated the utility of PET/MRI for the evaluation of CS [57,58,59,60]. Wisenberg et al. showed that PET/CT and PET/MRI provided similar diagnostic data for FDG uptake in a same-day protocol in 10 CS patients [58]. Comprehensive imaging with cardiac PET/MRI holds great promise for providing further insights regarding the disease process underlying CS. The combination of information on active inflammation from FDG-PET and fibrotic scars from LGE on MRI can provide incremental value in the development of therapeutic strategies for CS [57]. Wicks et al. assessed the diagnostic and prognostic utility of simultaneous cardiac PET/MRI in 51 consecutive patients with suspected CS. In their results, hybrid PET/MRI was superior to PET and MRI alone for detecting CS, and the presence of LGE and FDG uptake on PET/MRI was a predictor of major adverse cardiac events [59]. More recently, Guelich et al. demonstrated the utility of T1 and T2 mapping in addition to LGE in hybrid PET/MRI for the identification of active CS [60]. Thus, hybrid PET/MRI may have implications for the assessment of disease presence, activity, and prognosis in CS in a single examination.

### 4.3. Regional Myocardial Function

Cardiac involvement of sarcoidosis can show different manifestations ranging from early active granuloma to inactive scars in the myocardium [61]. This heterogeneity can also reflect the dyssynchrony of left ventricular wall motion. A recent meta-analysis of speckle-tracking echocardiography data demonstrated that left ventricular global longitudinal and circumferential strain, which represent the change in length in proportion to the baseline length from longitudinal and circumferential directions, was significantly lower in patients with extracardiac sarcoidosis but without any cardiac symptoms [62]. Assessment of global and regional ventricular function is one of the important functions of CMR. Accurate and reproducible measurements of left ventricular (LV) ejection fraction (EF) and LV volume are possible using CMR. Myocardial tissue-tracking imaging techniques, such as tagging, strain-encoded MRI, and myocardial feature-tracking deformation imaging, are useful for measuring regional myocardial contractile function (Figure 5). These CMR-derived strain parameters have the potential to facilitate early detection and prediction of disease outcome in patients with CS [63,64]. Velangi et al. reported that RV systolic dysfunction was present in 12.1% of 290 patients with sarcoidosis with known or suspected CS, and RV systolic dysfunction was independently associated with all-cause death [65] (Table 6).

PET-derived phase-analysis data can improve the sensitivity of scar detection in CS [66]. Phase analysis on gated PET [66] and single-photon emission computed tomography (SPECT) [67] images also has the potential to add incremental diagnostic or predictive value for patients with CS.

### 4.4. Computed Tomography

Coronary CT angiography is useful for excluding coronary artery disease in patients with suspected CS. Furthermore, delayed-phase contrast-enhanced CT, which is often performed 6–15 min after contrast injection, can be used to detect cardiac involvement in patients with systemic sarcoidosis in a manner similar to CMR (Figure 6) [5]. The iodine-based CT contrast material accumulates in the myocardial scar in a manner similar to gadolinium-based CMR contrast material [68]. Therefore, contrast-enhanced CT has the potential to serve as a “one-stop shop” for evaluation of coronary artery disease, systemic disease, and cardiac involvement in patients with sarcoidosis [69]. Delayed-enhanced CT can also be performed for patients with MRI-incompatible devices. We previously demonstrated that the image quality of delayed-enhanced CT is acceptable for evaluation of myocardial hyperenhancement in patients with or without implantable devices [5]. The disadvantages of contrast-enhanced CT are the relative contraindication to chronic kidney disease and the lower contrast-to-noise ratio in comparison with LGE-CMR [70]. Low tube voltage CT scanning with iterative model reconstruction [5] or virtual monochromatic imaging in dual-energy CT [71] may offer a more accurate assessment of myocardial hyperenhancement in this setting. Statistical texture analysis-based approaches have emerged as more objective and reproducible methods than visual assessment for detecting myocardial hyperenhancement in delayed-enhanced CT [72].

## 5. Conclusions

In this review, we have summarized the diagnostic approach for CS, including recent advances in PET, CMR, and CT. Current international recommendations for diagnosing CS are based on a multimodal approach (Figure 7). Among the existing modalities, FDG-PET and LGE evaluations on CMR are essential tools, and further advancements using novel techniques may be useful for the non-invasive diagnosis and prognosis assessment of CS.

## Figures and Tables

**Figure 1 jcm-10-05808-f001:**
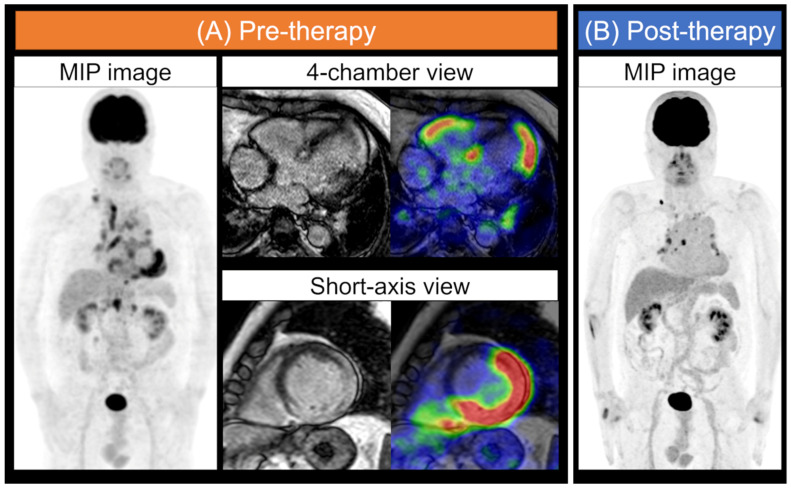
Typical FDG-PET and MR images for cardiac sarcoidosis. Cardiac sarcoidosis: A patient with systemic sarcoidosis was assessed for a cardiac lesion. Whole-body maximum intensity projection (MIP) image in FDG-PET obtained after an extended fasting period and consumption of a low-carbohydrate food preparations, and the four-chamber view and short-axis images of MRI LGE and a fused image of MRI and FDG-PET are displayed (**A**). LGE and FDG uptake in the LV and RV myocardium were detected. The physiological myocardial FDG uptake was suppressed. The MIP image also shows several regions of FDG uptake in the right supraclavicular, mediastinum, hilar, and abdominal lymph nodes. MIP images of post-anti-inflammatory therapy show that the uptake of some lymph nodes persisted, but abnormal cardiac accumulation was almost decreased (**B**). MIP, maximum intensity projection; FDG, ^18^F-fluorodeoxyglucose; PET, positron emission tomography; MRI, magnetic resonance imaging; LGE, late gadolinium enhancement; LV, left ventricular; RV, right ventricular.

**Figure 2 jcm-10-05808-f002:**
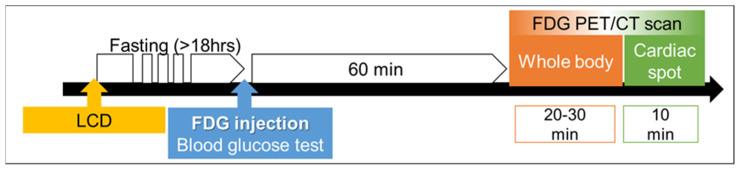
Typical time schedule of the FDG-PET/CT scan. Preparations before FDG-PET/CT include a low-carbohydrate diet (LCD) and prolonged fasting (over 18 h). Delayed cardiac spot imaging is also recommended if the time schedule permits it.

**Figure 3 jcm-10-05808-f003:**
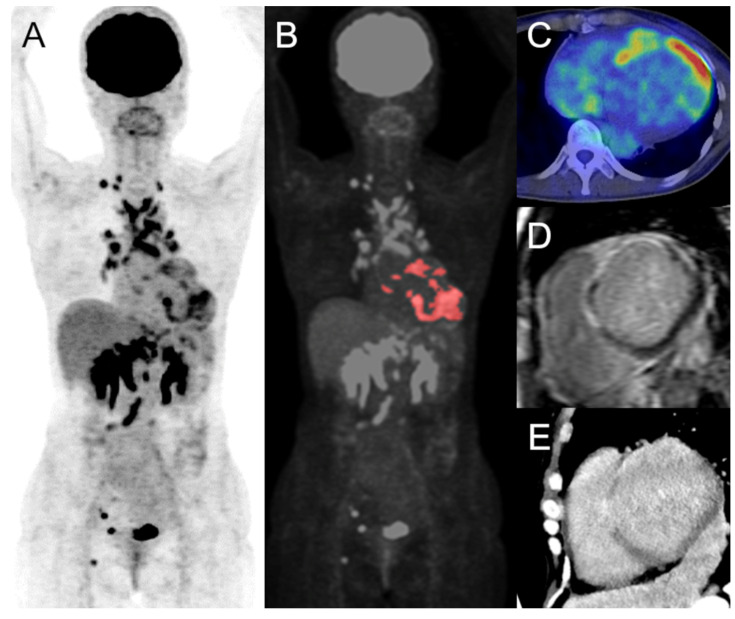
Volume-based analysis of FDG-PET. The maximum intensity projection (**A**), colored on the cardiac lesion (**B**), FDG-PET/CT of axial image (**C**), short-axis images of delayed-enhancement MRI (**D**), and CT (**E**) of cardiac sarcoidosis are displayed, respectively. There are focal uptakes in the left and right ventricles, which is concordant with the delayed enhancement on MRI and CT, indicating active cardiac sarcoidosis. The estimated SUVmax, cardiac metabolic volume, and cardiac metabolic activity were 10.8, 51.8, and 231.7 mL, respectively. There are also multiple FDG uptakes in the supraclavicular, mediastinum, hilum, abdomen, pelvis, and right inguinal lymph nodes. FDG, 18F-fluorodeoxyglucose; PET, positron emission tomography; CT, computed tomography; MRI, magnetic resonance imaging; SUV, standardized uptake value.

**Figure 4 jcm-10-05808-f004:**
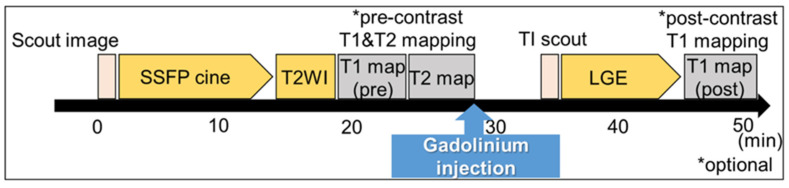
Typical time schedule of a CMR scan. Scout scans of the transaxial, coronal, and sagittal images were acquired. ECG-gated imaging is performed as follows: steady-state free precession (SSFP) short-axis and long-axis cine images are acquired during a breath-hold. T1 mapping images, which are optional, are obtained pre- and post-contrast administration. T2 mapping images are also optional. Late gadolinium enhancement (LGE) images are obtained from post-contrast images with an inversion time chosen to null normal myocardium. High-resolution gradient echo inversion recovery 10 min post-contrast.

**Figure 5 jcm-10-05808-f005:**
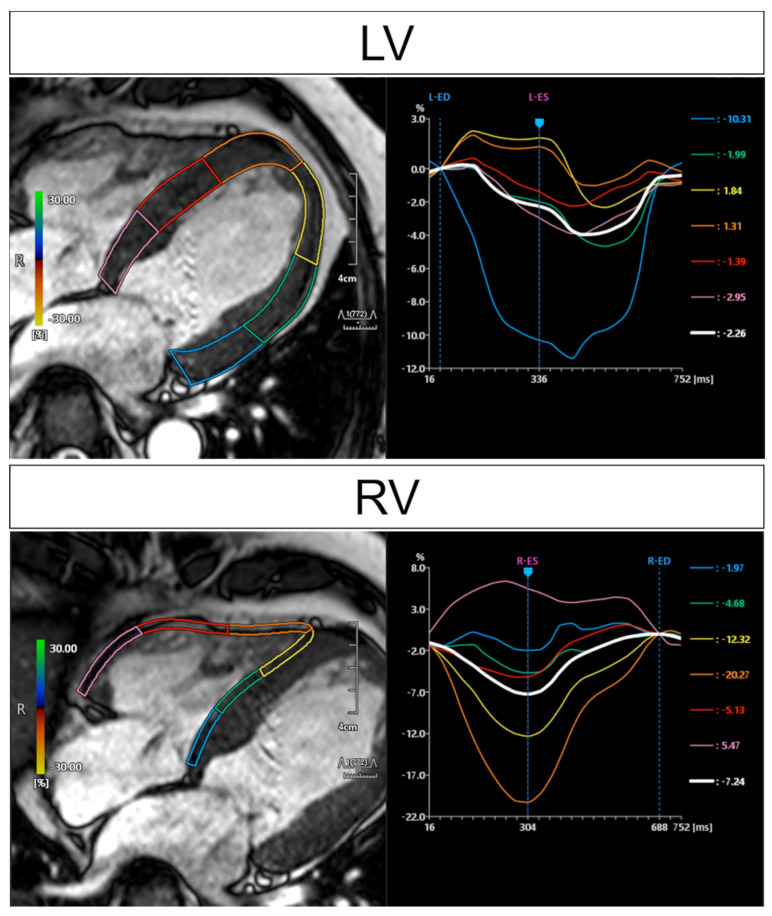
CMR feature-tracking analysis. Findings for the same patient as in Figure 1. Each regional strain is presented in the same colored curve, and the white curve represents the average of all regions. Inhomogeneous longitudinal strain due to cardiac sarcoidosis not only in the LV wall but also in the RV wall. LV, left ventricular; RV, right ventricular.

**Figure 6 jcm-10-05808-f006:**
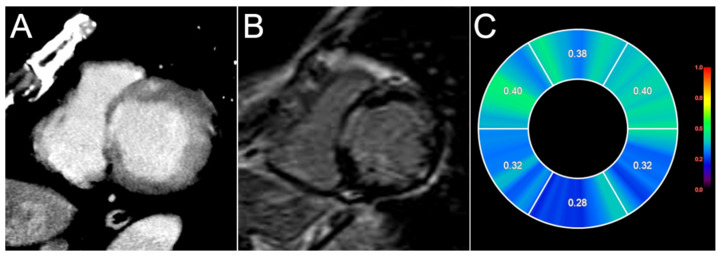
T1 mapping on MRI and delayed contrast-enhancement CT. A case of cardiac sarcoidosis. Short-axis views of delayed-enhancement CT (**A**), late gadolinium enhancement (LGE) on MRI (**B**), and the extracellular volume (ECV) map obtained pre- and post-enhancement at similar levels (**C**). Abnormal patchy enhancement in the anteroseptal, septal, and lateral wall was detected by delayed-enhancement CT. LGE can also reveal obvious hyperintensity areas consistent with CT. Higher ECV values corresponded to LGE.

**Figure 7 jcm-10-05808-f007:**
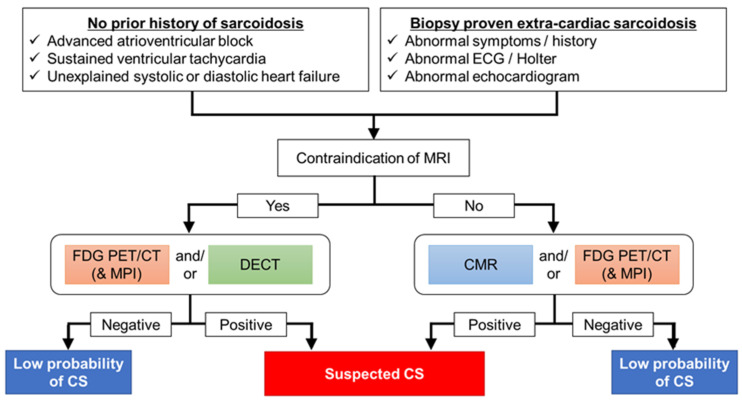
Diagnostic algorithm for a patient with suspected cardiac sarcoidosis. Both cardiac MRI and FDG-PET scans are effective in patients with suspected cardiac sarcoidosis. These scans are sometimes difficult to perform and interpret accurately unless the experts understand the potential pitfalls and artifacts. It is important to differentiate CS from other cardiac diseases such as coronary artery diseases, which might be false-positive for FDG-PET and CMR. Therefore, it is necessary to select the appropriate diagnostic protocol for each facility. FDG-PET should be considered to assess inflammatory activity before initiating anti-inflammatory therapy. In patients with implantable devices, PET would be the first choice. For detection of late enhancement, delayed iodine enhanced CT would be one option for the evaluation of coronary artery disease, systemic disease, and cardiac involvement in patients with sarcoidosis.

**Table 1 jcm-10-05808-t001:** The Japanese Society of Sarcoidosis and other Granulomatous Disorders (JSSOG) 2015 criteria for cardiac sarcoidosis [11,12].

1. Histological diagnosis group Cardiac sarcoidosis is confirmed when endomyocardial biopsy specimens demonstrate noncaseating epithelioid cell granulomas with a histological or clinical diagnosis of extracardiac sarcoidosis. 2. Clinical diagnosis group Cardiac sarcoidosis is confirmed when, despite the absence of noncaseating epithelioid cell granulomas on endomyocardial biopsy specimens, extracardiac sarcoidosis is diagnosed histologically or clinically and the following conditions and the following diagnostic criteria are satisfied: (a) Two or more of the five major criteria are met; OR (b) One of the five major criteria and two or more of the three minor criteria are met. Major criteria (1) Advanced atrioventricular block or sustained ventricular tachycardia. (2) Basal thinning of the interventricular septum or morphological abnormality (aneurysm, wall thinning, or wall thickening). (3) Depressed ejection fraction (<50%) or regional wall motion abnormality. (4) Abnormal uptake of ^67^Ga or ^18^F-fluorodeoxyglucose in the heart. (5) Delayed gadolinium enhancement on cardiac magnetic resonance. Minor criteria (1) Abnormal electrocardiographic findings: ventricular arrhythmias (non-sustained ventricular tachycardia or multifocal or frequent premature ventricular contractions), bundle branch block, axis deviation, or abnormal Q-waves. (2) Perfusion defects on nuclear imaging. (3) Endomyocardial biopsy: interstitial fibrosis or monocyte infiltration of moderate grade.

**Table 2 jcm-10-05808-t002:** Diagnosis of cardiac sarcoidosis from the HRS Expert Consensus Statement [9].

1. Histologic diagnosis Noncaseating granulomas on endomyocardial biopsy without an alternative cause 2. Clinical diagnosis (a) Histologic diagnosis of extracardiac sarcoidosis; and (b) The presence of one of the following findings: (1) Steroid-responsive cardiomyopathy or heart block. (2) Unexplained LVEF < 40%. (3) Unexplained sustained VT. (4) Advanced heart block. (5) Patchy uptake on cardiac PET. (6) Late gadolinium enhancement (LGE) on CMR. (7) Positive ^67^Ga uptake. and (c) Other causes for the cardiac manifestation(s) have been excluded.

**Table 3 jcm-10-05808-t003:** Advantages and disadvantages of FDG-PET and MRI.

	Advantages	Disadvantages
PET	Active lesion Whole-body imaging possible Assesses response to therapy Safely performed in patients with advanced renal dysfunction Intracardiac devices	Radiation exposure Lower spatial resolution Long acquisition time Need for specialized patient preparation Nondiagnostic scans due to physiological uptake More expensive
MRI	High spatial resolution Excellent soft-tissue contrast Non-ionizing radiation Detects morphological abnormalities including ventricular wall thinning A lower number of nondiagnostic scans No need for specialized patient preparation	Long acquisition time Limited by the incompatible cardiac devices With risk from gadolinium contrast in patients with advanced renal dysfunction

**Table 4 jcm-10-05808-t004:** Recommended foods for dietary preparation before FDG-PET.

Drink	Sugar-free drinking water such as tap water, mineral water, and tea, etc.
Vegetables	Low-carbohydrate vegetables such as broccoli, cabbage, cucumber, green pepper, lettuce, spinach, etc.
Low-carbohydrate food	Butter (without sugar), cheese, egg, fish, meat, tofu, etc.
Seasoning	Sugar-free seasoning, such as mayonnaise, olive oil, pepper, salad oil, etc.

**Table 5 jcm-10-05808-t005:** Diagnostic ability of the other tracers of PET.

PET Tracer	Reference	Gold Standard	Active CS/Total Number	Sensitivity (%)	Specificity (%)
DOTATOC	Lapa, C [33]	CMR	10/15	70.0	100.0
FMISO	Furuya, S [35]	FDG-PET	8/9	77.8	100.0
FLT	Norikane, T [36]	FDG-PET	13/20	92.3	100.0
	Martineau, P [37]	FDG-PET	7/14	85.7	100.0

CS, cardiac sarcoidosis; DOTATOC, ^68^Ga-DOTA-Tyr-octreotide; CMR, cardiac magnetic resonance imaging; ^18^F-fluoromisonidazole; FDG, ^18^F-fluolodeoxyglucose; PET, positron emission tomography; FLT, 3′-Deoxy-3-[^18^F]-fluorothymidine.

**Table 6 jcm-10-05808-t006:** Imaging evaluation relating to prognosis and/or cardiac events.

Modality	Authors	Findings
FDG-PET	Blankstein et al. [21]	Abnormal uptake and focal perfusion defects RV FDG uptake
	Ahmadian et al. [26]	CMA
	Sperry, B.W. et al. [29] Schildt, J.V. et al. [30]	Heterogeneity of myocardial FDG uptake
	Manabe, O. et al. [31]	Texture parameter
CMR	Greulich, S. et al. [48] Coleman, G.C. et al. [49]	Presence of LGE
	Ise, T. et al. [50] Kazmirczak, F. et al. [51]	Extent of LGE
	Dabir, D. et al. [64]	Global longitudinal strain
	Velangi, P.S. et al. [65]	RV systolic dysfunction

FDG, ^18^F-fluuorodeoxyglucose; PET, positron emission tomography; RV, right ventricular; CMA, cardiac metabolic response; CMR, cardiac magnetic resonance imaging; LGE, late gadolinium enhancement.

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
