# Peer review of "Advances in Diagnostic Imaging for Cardiac Sarcoidosis"

_jcm, 2021, doi:10.3390/jcm10245808_

Round 1
Reviewer 1 Report
The authors summarize cardiac imaging modalities for cardiac sarcoidosis, which is a relevant problem in this population. Overall, the paper gives a useful overview but should be amended regarding a couple of aspects:
My specific points are outlined below:
- in the introduction, it should be mentioned that the incidence and prevalence of CS differs regionally and depending on ethnicity (higher in japan than elsewhere); some data should be given.
- the current ERS clinical practice recommendation should be added since it recommends treatment options
- some important references are missing, such as PMID 34261086; 34132675; 27922863 to name some examples
- the authors should recommend more clearly what their preferred imaging modality is for suspected CS (I assume Echocardiography / CMR / PET?): what exactly do the authors recommend in patients with implantable devices (a common situation in CS)?
- The layout is a bit disorganized: figure 1: labeling of the panels missing.
- section 3.2 rename "other tracers": why bold?
- suggest adding an overview table summarizing the different imaging modalities compiling advantages, disadvantages, costs, diet requirements, etc.
- the authors could add their suggested protocol for evaluation of CS
- more details on diet for PET would be helpful for clinicians (which food to add, which to avoid?) patients and providers usually don't know exactly to what extent high fat low carb diet is required.
- a paragraph on imaging relating to prognosis / longitudinal evaluation would be helpful
Author Response
We appreciate your kind consideration of our paper and the constructive comments. The comments from the reviewers have been helpful in the revision of our manuscript. We have made extensive revisions to the manuscript in response to the reviewers’ comments and believe that it is greatly improved as a result. We hope you find that it is ready for publication. Please show the attached word file.

Reviewer 2 Report
The paper is interesting but requires several specifications.
For FDG-PET the problem of sensitivity and specificity for CS is not adequately addressed.
About FDG-PET it is not clear how specific for CS could be a positive result in a general referral population, even though after adequate diet preparation, in presence of previous myocardial infarction or chronic segmental ischemic heart disease or diabetes. How metabolic syndrome and obesity could affect the results ? The Authors should discuss these problems in more detail.
The sensitivity and specificity of the other tracers for CS (paragraph 3.2) is even less clear.
How many times a cardiac biopsy specimen was performed and was positive for CS after a positive pattern in a FDG-PET examination ? Or the diagnosis of CS in presence of a positive pattern in a FDG-PET examination could not be assessed prospectively, but only confirmed when FDG-PET pattern decreased after adequate treatment ?
For CMR (paragraph 4) the problem of specificity for CS is even worse since gadolinium can be found to be present in a variety of situations in a general referral population.
The same for integrated PET-CMR technique (why to call one time CMR and the other MRI ?)
The Authors should clarify if the reported studies and results are mainly dealing with a preselected population of suspected CS. It is in fact not clear if the diagnosis of suspected CS and the inclusion into the studies was done after confirmation of CS during follow-up, on the basis of the results of treatment and not blindly in an unselected population.
The Authors should then discuss and differentiate how all the techniques they are presenting are useful in a general unselected population to suspect the diagnosis of CS, or if those same techniques are mainly useful when the diagnosis of CS have to be made in a patient already known to have systemic sarcoidosis. In this case the technique could be useful mainly for prognosis and follow-up.
The discussion on how to select a patient and when indicate to perform one of the proposed techniques is then extremely important.
In the paragraphs 4.3 and 4.4 it should be clarified in more detail how to distinguish ischemic heart disease from CS in situations where CS is not present or when CS and ischemic heart disease are coexistant.
Author Response

(The authors gave the same response as above.)

Round 2
Reviewer 2 Report
The Authors have made the required changes.